# Tissue-resident memory T cells populate the human brain

Joost Smolders [1,2], Kirstin M. Heutinck[3], Nina L. Fransen [1], Ester B.M. Remmerswaal[3,4], Pleun Hombrink[5], Ineke J.M. ten Berge [3,4], René A.W. van Lier[5], Inge Huitinga [1] & Jörg Hamann[1,3]

Most tissues are populated by tissue-resident memory T cells ($T_{RM}$ cells), which are adapted to their niche and appear to be indispensable for local protection against pathogens. Here we show that human white matter-derived brain CD8$^+$ T cells can be subsetted into CD103$^-$CD69$^+$ and CD103$^+$CD69$^+$ T cells both with a phenotypic and transcription factor profile consistent with $T_{RM}$ cells. Specifically, CD103 expression in brain CD8$^+$ T cells correlates with reduced expression of differentiation markers, increased expression of tissue-homing chemokine receptors, intermediate and low expression of the transcription factors T-bet and eomes, increased expression of PD-1 and CTLA-4, and low expression of cytolytic enzymes with preserved polyfunctionality upon activation. Brain CD4$^+$ T cells also display $T_{RM}$ cell-associated markers but have low CD103 expression. We conclude that the human brain is surveilled by $T_{RM}$ cells, providing protection against neurotropic virus reactivation, whilst being under tight control of key immune checkpoint molecules.

[1] Department of Neuroimmunology, Netherlands Institute for Neuroscience, Meibergdreef 47, 1105BA Amsterdam, The Netherlands. [2] Department of Neurology, Canisius Wilhelmina Hospital, Weg door Jonkerbos 100, 6532SZ Nijmegen, The Netherlands. [3] Department of Experimental Immunology, Amsterdam Infection & Immunity Institute, Amsterdam UMC, University of Amsterdam, Meibergdreef 9, 1105AZ Amsterdam, The Netherlands. [4] Renal Transplant Unit, Department of Internal Medicine, Amsterdam Infection & Immunity Institute, Amsterdam UMC, University of Amsterdam, Meibergdreef 9, 1105AZ Amsterdam, The Netherlands. [5] Department of Hematopoiesis, Sanquin Research and Landsteiner Laboratory, Amsterdam Infection & Immunity Institute, Amsterdam UMC, University of Amsterdam, Meibergdreef 9, 1105AZ Amsterdam, The Netherlands. These authors contributed equally: Joost Smolders, Kirstin M. Heutinck, Nina L. Fransen. Correspondence and requests for materials should be addressed to J.S. (email: j.smolders@cwz.nl) or to J.H. (email: j.hamann@amc.uva.nl)

CD8[+] T cells have a critical role in immune protection against invading pathogens, in particular viruses. Upon infection, naive T lymphocytes are activated in secondary lymphoid organs and expand to large numbers. After clearance of the infection, some of these activated T cells differentiate into so-called memory T cells. Central memory T cells ($T_{CM}$ cells) circulate through the blood and the secondary lymphoid organs, which collect lymph fluid from the body's peripheral sites. Effector memory T cells ($T_{EM}$ cells) move between the blood and the spleen, and bear the ability to enter non-lymphoid tissues in case of an (re)infectious challenge. More recently, it became clear that tissues, which are common portals of reinfection, are populated by distinct lineages of tissue-resident memory T cells ($T_{RM}$ cells)[1–4]. $T_{RM}$ cells orchestrate the response to pathogens (re)encountered at these locations. Using the canonical markers CD69 and CD103, $T_{RM}$ cells have been identified in most murine and human tissues[5,6].

The central nervous system (CNS) is structurally and functionally unique but, in common with other tissues, requires efficient immune protection against infections[7]. This is illustrated by the ability of neuropathic viruses to enter the CNS and cause life-threatening infections[8]. The CNS is floating in cerebrospinal fluid (CSF), a functional equivalent of the lymph that is generated in the choroid plexus from arterial blood and reabsorbed into the venous blood at the arachnoid villi. The CSF contains CD4[+] and, to a lesser extent, CD8[+] T cells, which patrol the boarders of the CNS and provide protection[9]. These cells express CCR7, L-selectin, and CD27, indicating a $T_{CM}$-cell phenotype[10]. The parenchyma of the CNS was long believed to be an immune-privileged site, separated by tight cellular barriers from the blood and the CSF stream and, thus, being inaccessible for T cells. More lately, CD8[+] $T_{RM}$ cells have been identified in the parenchyma of the mouse CNS, where they provide local cytotoxic defense against viral infections[11–13].

We recently phenotyped human T cells acutely isolated from the post-mortem brain[14]. T cells in the corpus callosum had a CD8[+] predominance and were mostly located around blood vessels, presumably in the perivascular Virchow-Robin space. Their chemokine receptor profile lacked the lymph node-homing receptor CCR7, but included the tissue-homing receptors $CX_3CR1$ and CXCR3. The absence of the costimulatory molecules CD27 and CD28 suggested a differentiated phenotype[15,16], yet no perforin and little granzyme B were produced[14]. These cytotoxic effector molecules are characteristic for circulating effector-type CD8[+] T cells but lack in certain human $T_{RM}$-cell populations[17].

We here test the hypothesis that the CD8[+] T-cell compartment in the human brain harbors populations with $T_{RM}$-cell features and demonstrate the existence of two CD69[+] subsets, distinguished by the surface presence of CD103. We provide expression profiles of molecules associated with cellular differentiation, migration, effector functions, and transcriptional control in these cells, as well as cytokine profiles after stimulation. We propose that CD103 expression reflects antigen- and/or tissue compartment-specific features of these cells. Furthermore, we explore characteristics of the lesser abundant brain CD4[+] T-cell fraction and show that they are also enriched for $T_{RM}$ cell-associated surface markers, except for a notably low expression of CD103.

## Results

### Flow cytometry analysis of human brain T cells.
We designed multicolor flow cytometry panels to simultaneously assess T-cell phenotype, differentiation, activation, exhaustion, senescence, transcriptional regulation, homing characteristics, cytotoxic capacity, and cytokine production in brain isolates. Freshly isolated T cells of subcortical white matter and paired peripheral blood of deceased human brain donors were analyzed using these panels (Supplementary Figure 1). For comparison, we analyzed peripheral blood mononuclear cells (PBMCs) of healthy individuals. Blood from deceased donors showed a CD8[+] T-cell phenotype congruent with a more terminally differentiated stage, with a distribution profile of differentiation markers similar to living donors (Supplementary Figure 2). Despite the variable background of the brain donors, consisting of patients with Alzheimer's disease, Parkinson's disease, dementia, depression, multiple sclerosis, as well as controls with no known neurological disorders (Table 1), brain T cells display a remarkably consistent phenotype that differs significantly from circulating T cells.

### Human brain white matter contains CD4[+] and CD8[+] T cells.
We analyzed viable T lymphocytes from subcortical white matter with flow cytometry and observed both a CD8[+] and CD4[+] fraction (Fig. 1a). Approximately three times less CD4[+] T cells were retrieved from the tissue when compared to CD8[+] T cells (Fig. 1b). Since isolation procedures may create bias in T-cell proportions[18], numbers of CD4[+] and CD8[+] T cells in normal-appearing sub-cortical white matter sections were also quantified with immunohistochemistry (Fig. 1c–f). We observed a median of 0.89 (interquartile range 0.77–1.57) CD4[+] compared to 1.94 (1.75–2.17) CD8[+] T cells mm$^{-2}$ white matter (Fig. 1g). Both CD4[+] and CD8[+] T cells are mostly found in close relationship with blood vessels, and immunofluorescence staining with laminin[19] revealed the majority to reside in the perivascular space (Fig. 1h–l).

### Brain CD8[+] T cells bear $T_{RM}$ cell-associated surface markers.
Paired blood and brain-derived CD8[+] T cells were analyzed for distribution of differentiation markers by HSNE[20], revealing a segregated clustering of blood-derived and brain-derived CD8[+] T cells in the plot (Fig. 2a–d). As we reported previously[14], human brain CD8[+] T cells are differentiated CD45RA$^-$CD45R0$^+$ cells that barely express the co-stimulatory molecules CD27 and CD28, weakly bear the IL-7 receptor α-chain (IL-7Rα, CD127), and lack the lymph node-homing receptor CCR7 (Fig. 2e–j). Staining for the $T_{RM}$-cell markers CD69 and CD103 (αE integrin) revealed that these cells highly express CD69 (Fig. 2k, m). While CD69 expression can also indicate cellular activation, brain CD8[+] T cells do not express other markers commonly associated with T-cell activation, such as Ki-67 and HLA-DR/CD38 (Supplementary Figure 3). This suggests CD69 to reflect a $T_{RM}$-cell phenotype. About 40% of the CD8[+] T cells express CD103 (Fig. 2l, m), and co-expression analysis showed that CD69[+]CD103$^-$ and CD69[+]CD103[+] cells are the dominant CD8[+] T-cell populations in the white matter (Fig. 2n).

CD103 positivity in brain CD8[+] T cells correlates with a higher expression of the differentiation markers CD27, CCR7, and IL-7Rα but a lower expression of CD28 and CD45RA (Fig. 3a–f). In human skin, CD8[+] $T_{RM}$ cells are defined by expression of CD49a ($α_1β_1$ integrin)[21,22]. Brain CD8[+] T cells also express CD49a, with high levels on CD69[+]CD103$^-$ and even more CD69[+]CD103[+] cells (Fig. 3g, h). The expression of the integrin CD49d ($α_4β_1$ integrin), which circulating CD8[+] T cells employ to cross the blood brain barrier and enter the perivascular space[23], is slightly lower by brain when compared to blood CD8[+] T cells, and equal on brain CD8[+] T cells irrespective of the expression of CD69 and CD103 (Fig. 3i, j). Brain CD8[+] T cells do not express the adhesion G protein-coupled receptor GPR56, a surrogate marker for cytotoxic lymphocytes[24] (Fig. 3k, l). Expression of KLRG1 (killer cell lectin-like receptor subfamily G member 1) and CD57, two molecules found on terminally differentiated/senescent cells with low expression on CD8[+] $T_{RM}$ cells[22,25], is low in brain- compared to blood-derived CD8[+] T cells (Fig. 3m–p). Expression

**Table 1 Brain donor characteristics**

| NBB | Disease | Sex | Age | Cause of death | pH CSF | PMD |
|---|---|---|---|---|---|---|
| 94–325 | NO | F | 51 | Pneumonia | 7.05 | 7:40 |
| 95–106 | NO | M | 74 | Myocardial infarction | 6.75 | 8:00 |
| 99–051 | MS | F | 45 | Legal euthanasia | 6.62 | 10:55 |
| 10–103 | NO | F | 79 | Cardiac insufficiency | 6.30 | 10:30 |
| 11–044 | NO | M | 51 | Suicide | 7.05 | 7:45 |
| 11–072 | NO | F | 76 | Hepatic failure | 6.87 | 7:15 |
| 12–059 | NO | F | 78 | Pneumonia | 6.41 | 4:35 |
| 12–104 | NO | M | 79 | Legal euthanasia | 6.71 | 6:30 |
| 14–025 | FTD | M | 65 | End stage FTD-ALS | 6.34 | 05:45 |
| 14–026 | AD | F | 80 | Dehydration/respiratory tract infection | 6.36 | 04:50 |
| 14–030 | PD | M | 56 | Legal euthanasia | ND | 07:25 |
| 14–031 | AD | F | 90 | Cachexia/dehydration | 6.29 | 06:30 |
| 14–032 | PD | M | 83 | Cachexia | 6.38 | 05:10 |
| 14–035 | PD | M | 77 | Legal euthanasia | 6.69 | 04:48 |
| 14–038 | MS | F | 35 | Legal euthanasia | 6.37 | 10:20 |
| 14–039 | MS | F | 75 | Unknown | ND | 09:45 |
| 14–041 | BD | F | 79 | Renal insufficiency | 6.31 | 08:00 |
| 14–043 | NO | F | 60 | Metastasized mammary carcinoma | 6.58 | 08:10 |
| 14–045 | FTD | F | 68 | Unknown | 6.68 | 06:05 |
| 14–046 | AD | F | 85 | Cachexia/dehydration | 6.28 | 08:00 |
| 14–047 | AD | M | 68 | Unknown | 6.64 | 06:10 |
| 14–049 | AD | F | 78 | Cachexia/dehydration | 6.21 | 04:45 |
| 15–011 | MS | F | 57 | Legal euthanasia | 6.40 | 07:30 |
| 15–047 | MS | F | 50 | Legal euthanasia | 6.62 | 09:05 |
| 15–064 | MS | M | 50 | Legal euthanasia | 6.55 | 10:50 |
| 15–082 | MS | F | 47 | Pneumonia | 5.78 | 08:35 |

*AD* Alzheimer's disease, *Age* age at death in years, *BD* bipolar disorder, *CSF* cerebrospinal fluid, *F* female, *FTD* frontotemporal dementia, *M* male, *MS* multiple sclerosis, *NBB* Netherlands Brain Bank registration number, *ND* not determined, *NO* no brain disease, *PD*, Parkinson's disease, *PMD* post-mortem delay = time between death and the end of the autopsy in hours. *Legal euthanasia* euthanasia or physician-assisted suicide under the Termination of Life on Request and Assisted Suicide Act of 2002 in The Netherlands

of GPR56 and CD57 is similarly low on all cells (Fig. 3l, p), while expression of KLRG1 is lower on CD69+CD103+ when compared to CD69+CD103− and CD69−CD103− cells (Fig. 3n).

Since the anatomical localization of CD8+ T cells in the human brain parenchyma requires specific migratory properties, we measured the expression of chemokine receptors that have been implicated in tissue-homing. CD69+CD103+ CD8+ T cells are enriched for expression of CCR5, CXCR3, CXCR6, and CX$_3$CR1 (Fig. 4a–h). Expression of CXCR3 is critical for the establishment of CD103+ T$_{RM}$ cells in the murine skin[25], and CXCR6 has been identified as part of the core phenotypic profile of T$_{RM}$ cells[22]. Overall, expression of CX$_3$CR1 and CCR5 are higher in brain CD8+ T cells compared to blood CD8+ T cells. To further test the assumption of a parenchymal localization of CD69+CD103+ CD8+ T cells, we immunohistochemically stained white matter of two brain donors for CD103. As expected, CD103+ T cells localize in the brain parenchyma but also, among CD103− T cells, in the perivascular space (Fig. 4i, j; Supplementary Figure 4). Therefore, CD103 positivity is no exclusive marker for localization in the brain parenchyma but may reflect a brain CD8+ T-cell subset with a greater propensity to migrate to (inflamed) parenchyma.

**Brain CD8+ T cells express T$_{RM}$-cell transcription factors.** Control of effector and memory CD8+ T-cell differentiation critically depends on the balance between the T-box transcription factors T-bet and eomes (eomesodermin)[26]. We found that most CD8+ T cells eluted from the brain express intermediate amounts of T-bet and low amounts of eomes (Fig. 5a, b)[15]. A down-regulated expression of eomes has been previously described as a characteristic feature of CD103+ T$_{RM}$ cells[25,27], with a down-regulation of T-bet expression being mandatory for the development of herpes simplex virus-specific CD103+ T$_{RM}$ cells in the murine skin[27]. Likewise, CD103 expression correlates with a low

expression of eomes in brain CD8+ T cells (Fig. 5c), and the proportion of T-bet-intermediate/eomes-low cells is highest in the CD69+CD103+ cell fraction (Fig. 5d). Hobit (homolog of Blimp-1 in T cells), which is a critical component of murine T$_{RM}$-cell transcriptional program[28], yet absent in human lung T$_{RM}$ cells[29], is not expressed by brain CD8+ T cells (Fig. 5e, f).

**Distinct effector molecules of CD103+ brain CD8+ T cells.** We next analyzed the production of cytolytic effector molecules by brain CD8+ T cells directly ex situ. In accordance with our previous study[14], brain CD8+ T cells show a low expression of granzyme B compared to peripheral blood and virtually no expression of the lytic mediator perforin (Fig. 6a–d). In contrast, granzyme K is expressed by comparable proportions of CD8+ T cells from blood and brain (Fig. 6e–g). Both granzyme B and K are predominantly produced by CD69−CD103− and CD69+CD103− T cells, with CD69+CD103+ cells showing a clearly smaller positive population for both enzymes (Fig. 6b, f).

Next, we analyzed cytokines expressed by brain CD8+ T cells upon activation. Almost all brain CD8+ T cells produce IFN-γ (interferon gamma) and TNF-α (tumor necrosis factor alpha) upon stimulation with PMA/ionomycin (Fig. 6h–l), with the majority being positive for both (Fig. 6r). CD8+ T cells positive for GM-CSF (granulocyte-macrophage colony-stimulating factor) and IL-17A (interleukin-17A) have been associated with neuroinflammation of multiple sclerosis[30]. A subset of brain CD8+ T cells produce GM-CSF (Fig. 6m–o), almost exclusively in addition to IFN-γ and TNF-α (Fig. 6r). We could detect only low proportions IL-17A+ brain CD8+ T cells (Fig. 6p, q). Expression of CD103 is not affected by stimulation with PMA/ionomycin (Fig. 6s). Despite a similar production of IFN-γ and TNF-α, slightly more CD103+ CD8+ brain T cells produce GM-CSF when compared to CD103− cells (Fig. 6o). Although being attributed pleiotropic functions, GM-CSF is predominantly

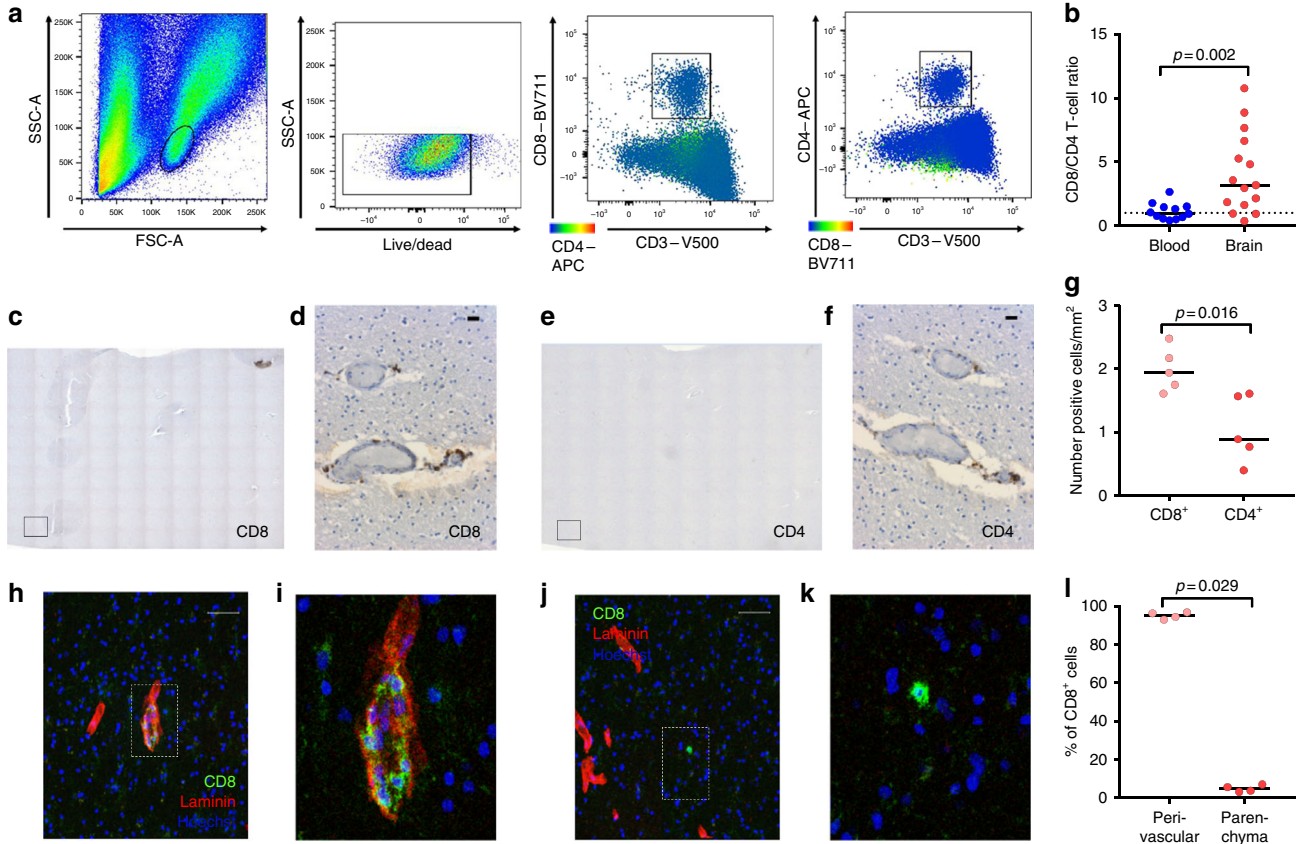

**Fig. 1** CD8[+] and CD4[+] T cells populate the human brain. **a** Gating procedure applied to analyze brain CD3[+]CD4[+] and CD3[+]CD8[+] T cells, eluted from normal-appearing white matter. **b** Quantification of the CD8[+]/CD4[+] T-cell ratio. Immunohistochemical staining of CD8- (**c**, **d**) and CD4- (**e**, **f**) immunoreactive cells in normal subcortical white matter of a donor without brain disease. **c**, **e** Overview of 10 × 10 tiled images at 10× magnification; the marked square indicates a bright field. **d**, **f** 20× magnification (scale bar = 20 μm). **g** Quantification of CD8-immunoreactive and CD4-immunoreactive cells (number/mm$^2$). **h–k** Immunofluorescent staining of CD8 (green), laminin (red), and Hoechst (blue) at 10× magnification (scale bar = 50 μm) (**h**, **j**) and a zoom-in (**i**, **k**). **l** Quantification of CD8-immunoreactive cells co-localizing with laminin (perivascular space) or not co-localizing with laminin (parenchyma). Bars show median values. *p*-values show Mann–Whitney *U* test

known as an important activation signal for monocytes/macrophages[31,32]. Therefore, despite producing less cytolytic enzymes directly ex situ, CD103[+] brain CD8[+] T cells display after stimulation a polyfunctional cytokine profile (Fig. 6t) capable to activate myeloid cells.

**Brain CD8[+] T cells express PD-1 and CTLA-4.** Since granzyme B is highly neurotoxic[33], we hypothesized the levels to be suppressed by signals from the brain microenvironment and analyzed the expression of inhibitory molecules. PD-1 (programmed death-1), a central regulator that preserves CD8[+] T cells from overstimulation, excessive proliferation, and terminal differentiation[34], is highly expressed in brain CD8[+] T cells, most prominently in the CD69[+]CD103[+] subset (Fig. 7a–c). Expression of CTLA-4 (cytotoxic T lymphocyte-associated antigen 4), a homolog of CD28, which blocks T-cell activation[35], is even more highly expressed in brain CD8[+] T cells, again most prominently in the CD69[+]CD103[+] subset (Fig. 7d–f). This is in line with the high CTLA-4 and PD-1 expression found in the core phenotypic signature of CD103[+] T$_{RM}$ cells[22,25]. Systemic treatment with soluble CTLA-4 has been associated with a suppression of CD8[+] T-cell granzyme B production in mice infected with lymphocytic choriomeningitis virus[36]. Conversely, in splenocytes of acute Friend retrovirus-infected mice, virus-specific PD-1-positive CD8[+] T cells produce the largest amounts of granzyme B[37]. The availability of the ligands for PD-1 or CTLA-4 may

determine their effect on granzyme expression. We performed immunohistochemical stainings of normal white matter of two brain donors for CD86 (ligand of CTLA-4) and PD-L1 (ligand of PD-1) and found no immunostaining for either one (Fig. 7g, h). To assess the effect of inflammation on CD86 and PD-L1 expression, we stained an HLA-DR-positive active demyelinating MS lesion and found microglia-like cells to stain positive for CD86 (Fig. 7i) and reactive astrocyte-like cells to stain positive for PD-L1 (Fig. 7j). These findings suggest that CTLA-4 and PD-1 may provide important inhibitory signals to brain CD8[+] T cells in inflammatory conditions.

**Brain CD4[+] T$_{RM}$ cells express low levels of CD103.** In accordance with their earlier described T$_{EM}$-like phenotype[14], brain CD4[+] T cells are high in expression of CD45R0 and show a low expression of CD27, CD45RA, CCR7, and CD28 (Supplementary Figure 5a–f). In peripheral blood, CD4[+] T cells with low CD27, CD28, and CCR7 expression associate with viral infections and display cytotoxic functions[38,39]. However, production of lytic mediators granzyme B and perforin by brain CD4[+] T cells is low, while granzyme K is expressed at higher levels (Supplementary Figure 5g–i). Likewise their CD8[+] counterparts, brain-derived CD4[+] T cells show a high CD69 expression (Fig. 8a). The CD69[+] CD4[+] T-cell fraction is not enriched for activation markers Ki-67, IL-2 receptor alpha-chain (IL-2Rα, CD25), or HLA-DR/CD38 co-expression (Supplementary Figure 6). In contrast to brain CD8[+]

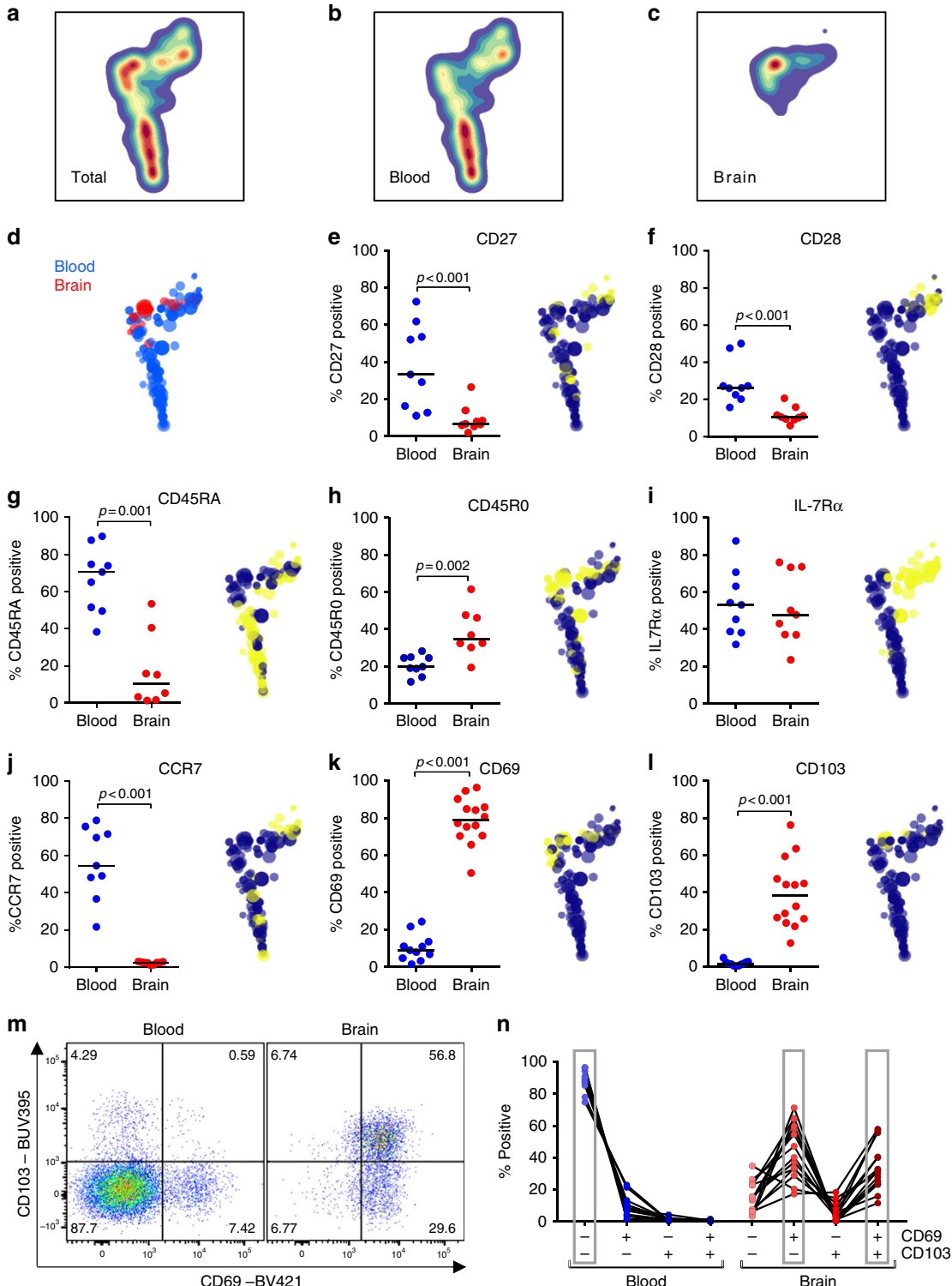

**Fig. 2** Human brain CD8+ T cells express the tissue residence markers CD69 and CD103. **a–c** HSNE plot of paired brain-derived and blood-derived CD8+ T cells of $n = 5$ donors, based on expression of markers shown in this figure, as well as KLRG1 and GPR56, shows segregated clustering of blood-derived and brain-derived CD8+ T cells. **d** Distribution of hierarchical clusters in the HSNE plot with the size of the dots indicating hierarchical cluster size. **e–l** Quantification of CD8+ T cells expressing CD27, CD28, CD45RA, CD45R0, IL-7Rα, CCR7, CD69, and CD103, respectively. In the HSNE plots, yellow dots indicate positive and blue dots negative hierarchical clusters. Clustering of brain CD8+ T cells is most prominently characterized by high expression of CD69 and CD103. Bars show median values. $p$-values show Mann–Whitney $U$ test; no brackets indicate no significant difference. **m** Dot plot of CD69 and CD103 co-expression in CD3+CD8+ T cells eluted from blood and brain. **n** Co-expression of CD69 and CD103. The dominant phenotype was CD69−CD103− in blood and CD69+CD103− and CD69+CD103+ in brain

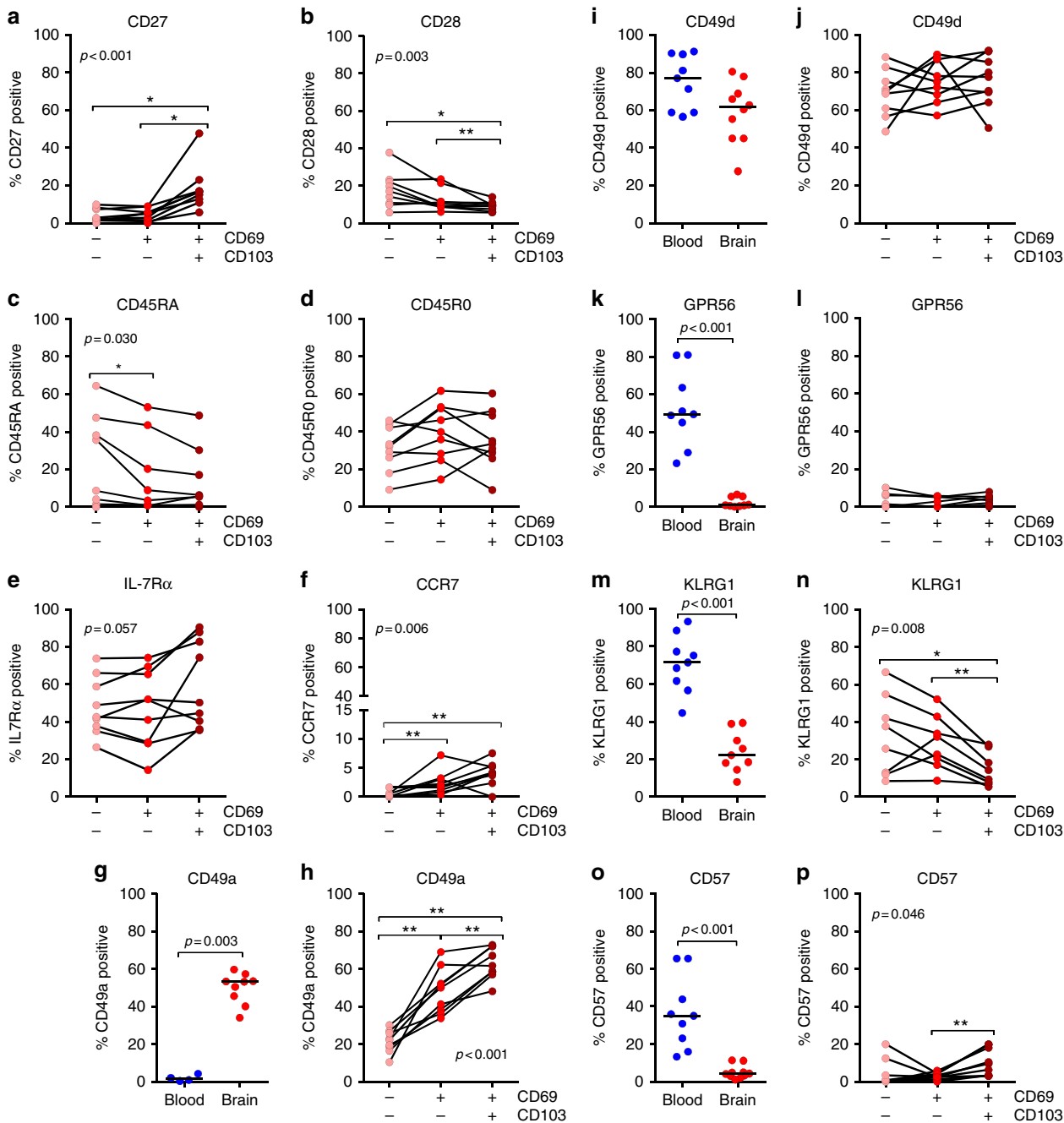

**Fig. 3** Human brain CD8[+] T cells distinctly express surface markers, based on CD69 and CD103 co-expression. **a–p** Quantification of CD8[+] T cells expressing CD27, CD28, CD45RA, CD45R0, IL-7Rα, CCR7, CD49a, CD49d, GPR56, KLRG1, and CD57, respectively, in $n = 9$ donors. Bars show median values. *p*-values show Mann–Whitney *U* test for unpaired data (**g**, **i**, **k**, **m**, **o**) and Friedman test for paired data with Wilcoxon signed ranks as post hoc test (**a–f**, **h**, **j**, **l**, **n**, **p**) (**p* < 0.05, ***p* < 0.01); no brackets indicate no significant difference

T cells, CD103 is sparsely expressed on brain-derived CD4[+] T cells (Fig. 8b–d). A restricted expression of CD103 on CD4[+] $T_{RM}$ cells has previously been described in several tissues[22,40,41].

We analyzed the expression of several other $T_{RM}$-cell phenotypic markers on CD69[−] and CD69[+] brain CD4[+] T cells (Fig. 8e–l). CD69[+] CD4[+] cells are enriched for expression of CD49a, PD-1, and CXCR6, which have been identified as the core-signature of CD4[+] $T_{RM}$ cells in multiple tissues[22]. Additionally, a higher expression of CTLA-4, CCR5, and CXCR3 is observed, which was earlier reported in human lung CD69[+]CD103[+/−] CD4[+] $T_{RM}$ cells[29]. A low expression of CX3CR1 has been associated with a CD4[+] $T_{RM}$-cell phenotype[22].

However, the CD4[+] T cells we isolated rather display an enrichment for CX3CR1, as we observe for brain CD8[+] T cells. CD4[+] CD69[+] T cells are not enriched for CD49d, which activated CD4[+] T cells use to migrate through an inflamed blood brain barrier[23].

## Discussion

Presented here is a phenotypic and functional profile of T cells in the human brain. Among CD8[+] cells, expression of differentiation markers, integrins, chemokine receptors, transcription factors, granzymes, cytokines, and immune checkpoint molecules

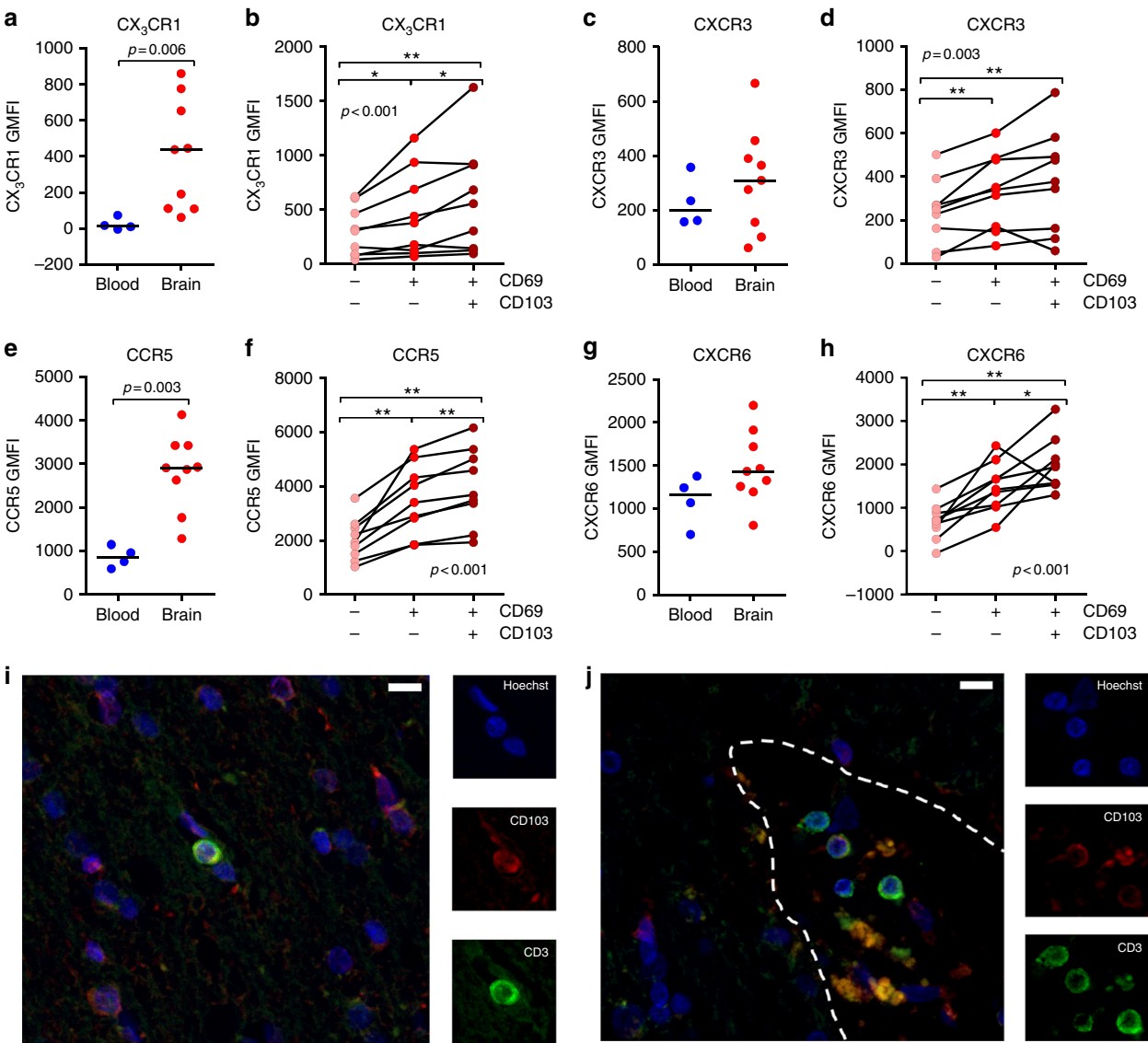

**Fig. 4** Human brain CD69[+]CD103[+] CD8[+] T cells are enriched for tissue-homing chemokine receptors. **a–h** Quantification of CD8[+] T-cell expression levels of CX$_3$CR1, CXCR3, CCR5, and CXCR6 (GMFI, geometric mean fluorescence intensity). Bars show median values. $p$-values show Mann–Whitney $U$ test for unpaired data (**a, c, e, g**) and Friedman test for paired data with Wilcoxon signed ranks as post hoc test (**b, d, f, h**) (*$p < 0.05$, **$p < 0.01$; no brackets indicate no significant difference. **i, j** Immunofluorescent staining for CD3 and CD103 of paraffin tissue shows a parenchymal (**i**) and perivascular (**j**) localization of CD3 (green) and CD103 (red) immunoreactive cells. Borders of the perivascular space were designated based on histological hallmarks (i.e., lymphocytes in close relationship with the extraluminal side of a blood vessel) and are marked with a dotted white line (scale bar = 10 μm)

revealed a profile that matches the core phenotypic and transcriptional signature of T$_{RM}$ cells, including the presence of a CD103[+]CD69[+] and a CD103[−]CD69[+] sub-population. Therefore, we further consider these cells as brain CD8[+] T$_{RM}$ cells. About half of all brain CD8[+] CD69[+] T$_{RM}$ cells express CD103. CD8[+] CD69[+] T$_{RM}$ cells that express CD103 show increased expression of chemokine receptors (CX$_3$CR1, CXCR3, CCR5, CXCR6), reduced production of cytolytic enzymes (granzyme B and K), and increased presence of inhibitory receptors (CTLA-4 and PD-1), compared to cells that lack CD103. This profile suggests a greater propensity of migration from the perivascular space into the brain parenchyma. CD4[+] CD69[+] T cells are also enriched for T$_{RM}$ cell-associated integrins, chemokine receptors, and inhibitory molecules, except for CD103. While the finding of phenotypically distinct CD69[+] T$_{RM}$-cell subsets provides a new perspective to the immune surveillance of the human CNS, we do at present only poorly understand the drivers of these diverse

phenotypes. Unravelling their kinetics and regulation may provide tools to modulate brain T-cell behavior for the benefit of patients suffering from infectious, inflammatory, or neoplastic conditions in the CNS.

CD69-expressing CD8[+] T$_{RM}$-cell subsets differ across human tissues; in the colon and the lung, most of these cells co-express CD103[5]. The equal distribution between CD103[−] and CD103[+] CD8[+]CD69[+] T$_{RM}$ cells that we found in the human brain is also seen in the mouse[13]. We speculate that anatomic compartmentalization may be a contributing factor. The periventricular space is an important immuno-active compartment between the blood–brain barrier and the brain parenchyma, were activated T cells and antigen-presenting cells reside and interact[7]. This is a different, immunologically vibrant microenvironment, compared to the brain parenchyma, where extensive immune activation can be harmful for neurons with a limited regenerative capacity. Mediators of granule exocytosis-mediated cytotoxicity, such as

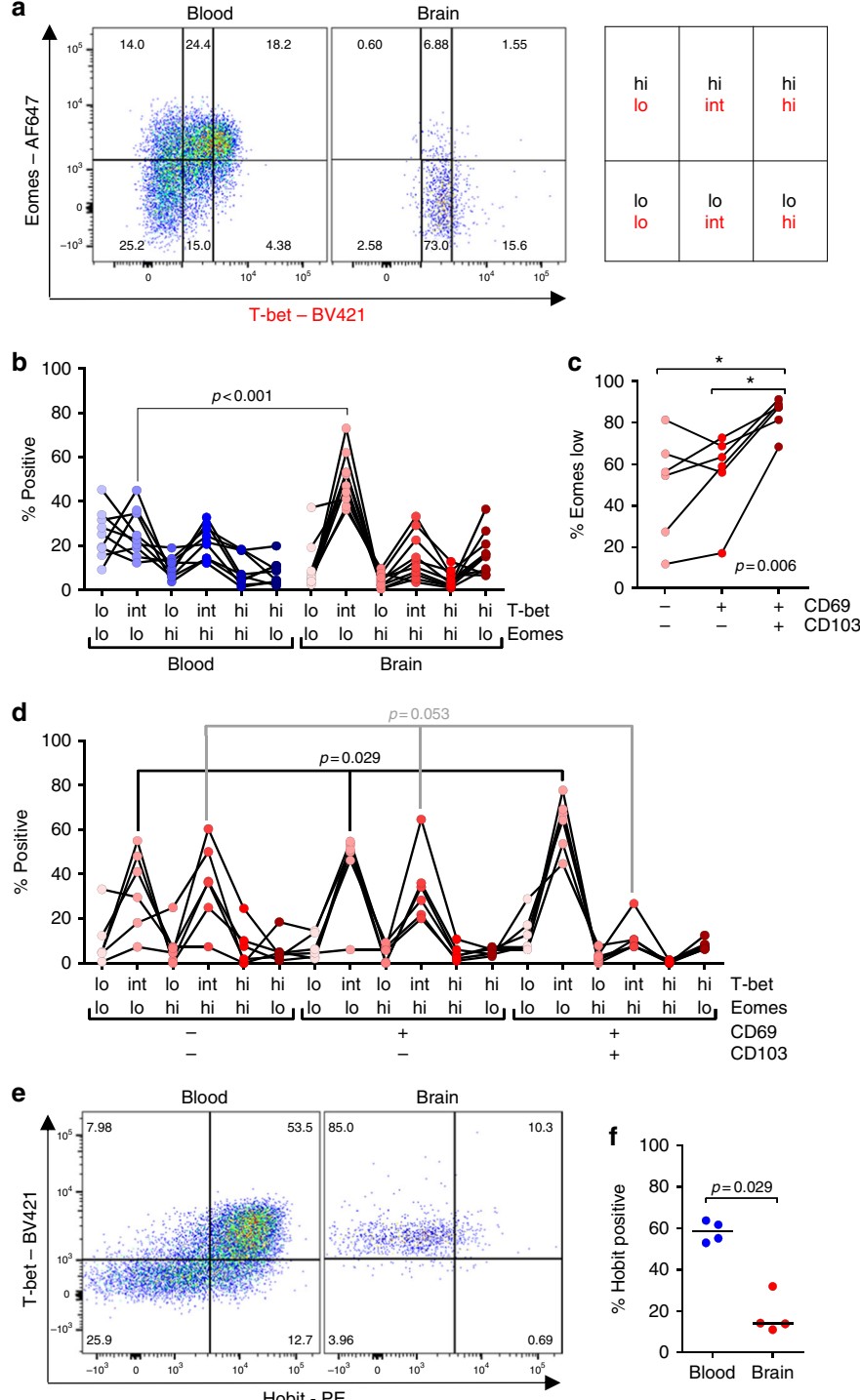

**Fig. 5** Human brain CD8[+] T cells show a T-bet-intermediate/eomes-low phenotype. **a** Dot plot showing the gating strategy T-bet and eomes in a paired blood and brain sample. T-bet and eomes co-expression (lo/lo > int/lo > lo/hi > int/hi > hi/hi > hi/lo) correlates in virus-specific CD8[+] T-cell with differentiation from central memory to terminally differentiated effector cells, respectively[15]. **b** Comparison between blood and brain CD8[+] T_RM cells, and stratification based on CD69 and CD103 co-expression of brain eomes low (**c**) and all T-bet/eomes subsets (**d**). **e** Dot plot of T-bet and Hobit co-expression by CD8[+] T cells from brain and blood of a donor. **f** Quantification of Hobit-positive cells is shown. Bars show median values. *p*-values show Mann–Whitney *U* test (**b**, **f**) and Friedman test with Wilcoxon signed ranks as post hoc test (**c**, **d**) (*$p < 0.05$, **$p < 0.01$); no brackets indicate no significant difference

perforin, granzyme A, and granzyme B, are highly neuro-toxic[33,42,43]. Release of these lytic enzymes should be under tight control, whilst maintaining the capability to elicit a fast inflammatory response when a neurotropic virus threatens the CNS. This profound inflammatory potential of brain CD8[+] is

highlighted by the substantial production of IFN-γ, TNF-α, and even GM-CSF upon activation. The polyfunctional inflammatory cytokine profile of human T_RM cells was previously also observed for lung CD103[+] and CD103[−] T_RM cells[44], and mouse cytomegalovirus-specific and *Toxoplasma gondii*-specific CD8[+]

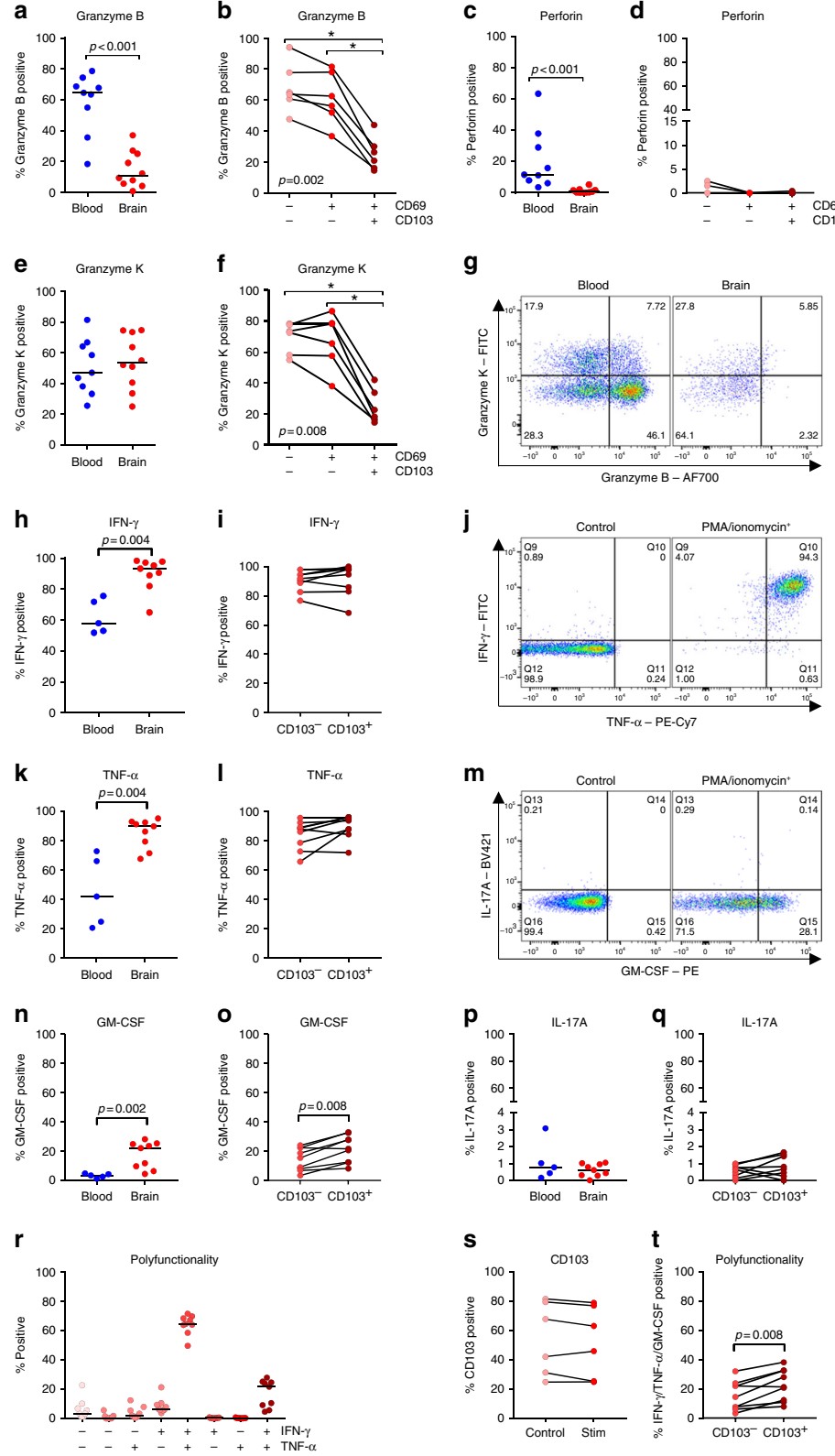

**Fig. 6** CD103+ brain CD8+ T cells express few cytolytic enzymes but show a polyfunctional cytokine profile. **a–f** Quantification of the percentage of brain CD8+ T cells directly ex situ expressing granzyme B, perforin, and granzyme K, respectively. **g** Representative dot plots of CD3+CD8+ T cells stained for granzyme B and granzyme K. **h–i**, **k–l**, **n–q** Quantification of the percentage of brain CD8+ T cells positive for IFN-γ, TNF-α, GM-CSF, and IL-17A after stimulation with PMA/ionomycin in vitro. **j**, **m** Dot plot of PMA/ionomycin-stimulated CD3+CD8+ T cells stained for the respective cytokines. **r** Quantification of IFN-γ, TNF-α, and GM-CSF co-expression, **t** stratified for expression CD103. **s** CD103 expression in brain CD8+ T$_{RM}$ control and PMA/ionomycin-stimulated cells. p-values show Mann–Whitney U test (**a**, **c**, **e**, **h**, **k**, **n**, **p**), Friedman test for paired data with Wilcoxon signed ranks as post hoc test (**b**, **d**, **f**), or Wilcoxon signed ranks test (**i**, **l**, **o**, **q**, **s**, **t**) (*p < 0.05, **p < 0.01; no brackets indicate no significant difference

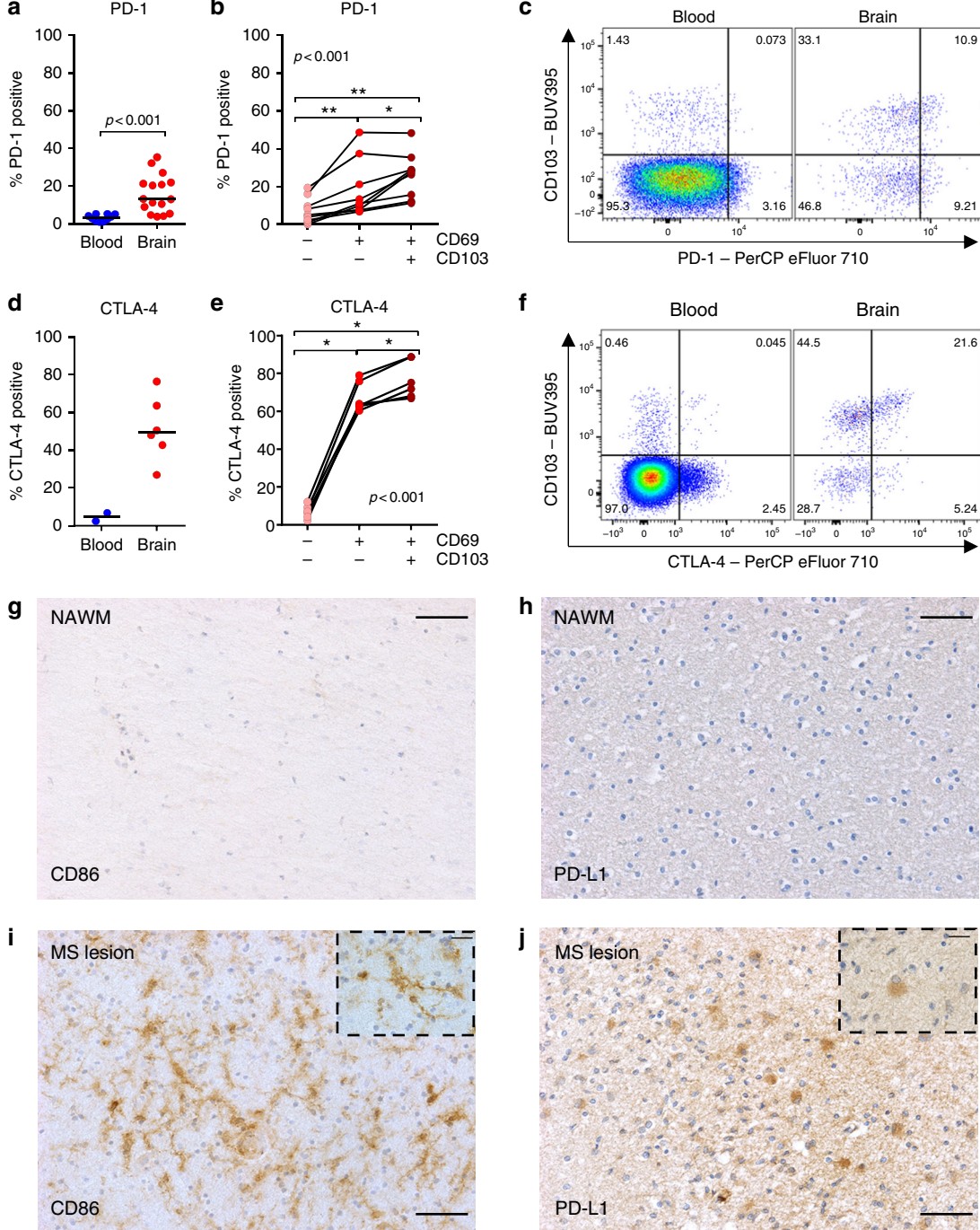

**Fig. 7** Enrichment CTLA-4 and PD-1 expression on brain CD8$^+$ T cells. **a**, **b**, **d**, **e** Quantification of the percentage of CD8$^+$ T cells expressing PD-1 and CTLA-4, respectively. Bars show median values. *p*-values show Mann–Whitney *U* test for unpaired data (**a**, **d**) and Friedman test for paired data with Wilcoxon signed ranks as post hoc test (**b**, **e**) (\**p* < 0.05, \*\**p* < 0.01); no brackets indicate no significant difference. Representative dot plots of CD3$^+$CD8$^+$ lymphocytes stained for **c** PD-1 and CD103 and **f** CTLA-4 and CD103. Immunohistochemical staining for **g** CD86 (ligand of CTLA-4, brown) and **h** PD-L1 (ligand of PD-1, brown) in a donor with Alzheimer's disease and a donor without brain disease showed no staining (scale bar = 50 μm). However, in an HLA-DR-positive active, demyelinating MS lesion of a donor with MS, specific staining for **i** CD86 and **j** PD-L1 was found in microglia and astrocyte-like cells, respectively (scale bar = 50 μm; scale bar insert = 20 μm)

brain T$_{RM}$ cells also produced IFN-γ and TNF-α[45,46]. GM-CSF may be of particular relevance to T$_{RM}$ cells residing in the white matter, since microglia express the GM-CSF-receptor complex and are readily activated by GM-CSF[47]. GM-CSF-immunoreactive CD8$^+$ T cells have also been observed in the context of MS white matter lesions[30].

Besides anatomic localization, functional differences implicate that antigen exposure may also underlie the distinct features of

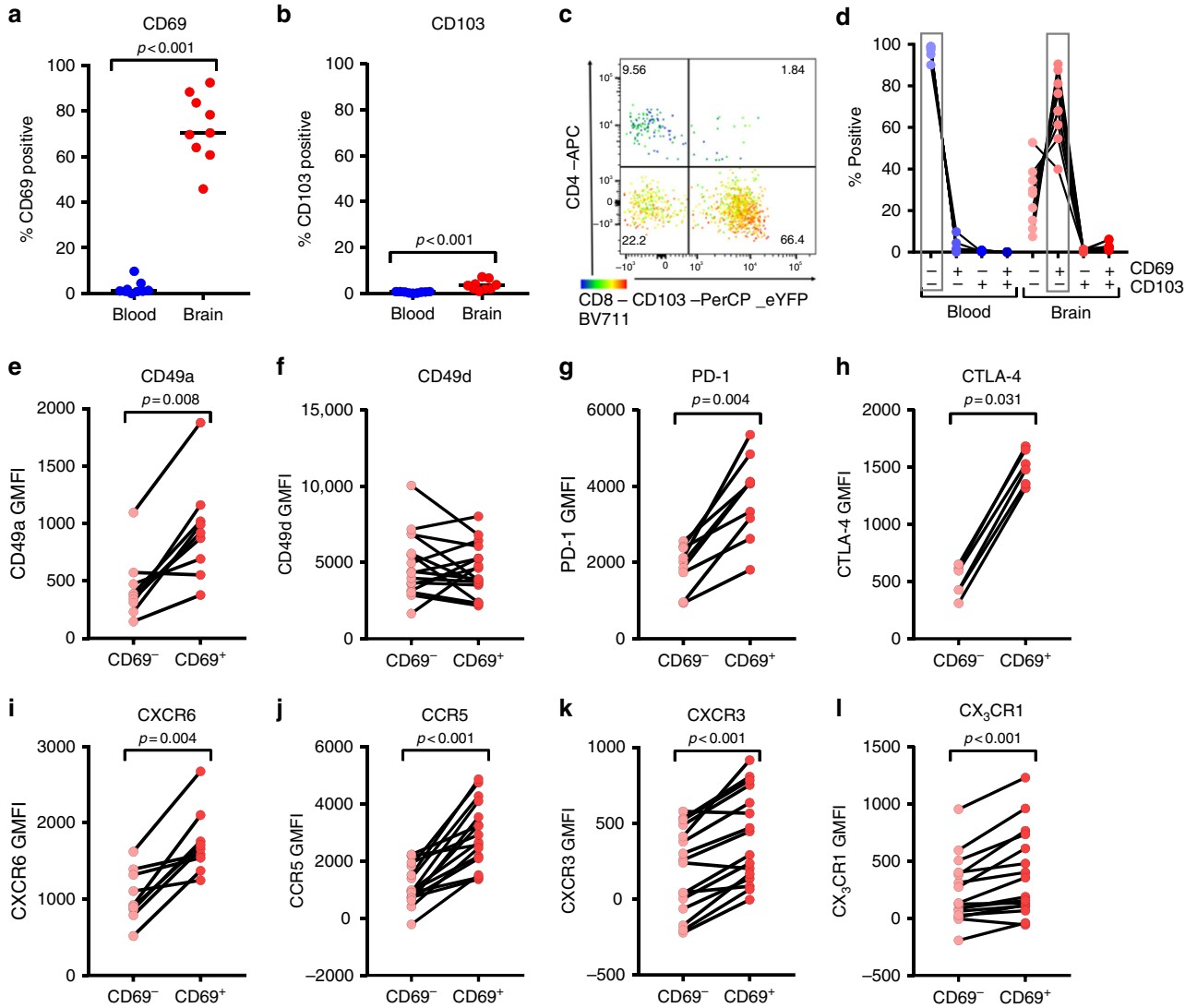

**Fig. 8** Human brain CD4+ CD69+ T cells are enriched for core phenotypic $T_{RM}$-cell markers. **a**, **b**, **d** Quantification of CD69 and CD103 (co-)expression. *p*-values show Mann–Whitney *U* test. **c** Representative dot plot of CD4, CD8, and CD103 staining of brain CD3+ CD69+ T cells. **d** Co-expression of CD69 and CD103. The dominant phenotype was CD69−CD103− in blood and CD69+CD103− in brain. **e–l** Quantification of CD4+ T-cell expression levels of CD49a, CD49d, PD-1, CTLA-4, CXCR6, CCR5, CXCR3, and CX3CR1 (GMFI, geometric mean fluorescence intensity). *p*-values show the Wilcoxon signed ranks test

CD103− and CD103+ CD8+ $T_{RM}$ cells. We report a consistent profile of surface markers and transcription factors, which points to an earlier phase of differentiation in CD103+ compared to CD103− CD8+ $T_{RM}$ cells. The exact antigens against which these cells are directed are at present poorly defined but do most likely comprise neurotropic viruses[9]. Notably, resident CD8+ T cells appear after herpes simplex virus infection in sensory ganglia and trigeminal ganglia, preventing reactivation in latent infection[48,49]. In mice, CD103+ CD8+ $T_{RM}$ cells appeared in the CNS after a vesicular stomatitis virus encephalitis[12] and chronic *Toxoplasma gondii* infection[46], and congenital infection with mouse cytomegalovirus induced a CD103+ and CD103− CD8+ $T_{RM}$-cell population in the brain parenchyma[45]. Contrastingly, autologous Epstein-Barr virus-infected B cell-reactive CD8+ T cells isolated from white matter lesions of MS patients did not show immunostaining for CD103 in situ[50]. Therefore, history of primary infection may also shape the composition of the brain CD8+ $T_{RM}$-cell pool. The reactivity of these cells to antigens of these and other viruses remains to be determined. In addition to specificity,

also the strength of antigenic stimulation during recruitment of CD8+ T cells may influence the composition of brain $T_{RM}$ cells[51].

Of note, PD-1, an immunological checkpoint for T-cell differentiation, is prominently expressed, with highest levels on CD103+ cells. Presence of PD-1 on the majority of CD8+ $T_{RM}$ cells has been described for other tissues, including eye, skin, lung, liver, and kidney[5,17,22,52,53], and may be a general characteristic of the transcriptional program of $T_{RM}$ cells. Whether PD-1 has a critical role in restricting the cytotoxic capacity and cytokine production of brain CD8+ $T_{RM}$ cells, thereby controlling excessive immunopathology, remains to be shown. A case of white matter T-cell infiltration with demyelination and macrophage activation in a patient after four courses of the PD-1 immune checkpoint inhibitor nivolumab has been reported[54]. Cases of bilateral, internuclear ophthalmoplegia and cerebellar ataxia have also been reported in PD-1 inhibitor-treated patients, without the neuroradiological substrate being specified[55]. Of note, human CNS cells express low levels of PD-L1 and PD-L2, the ligands of PD-1, under basal conditions but upregulate PD-L1

under inflammatory conditions[56]. Likewise, we find no immunostaining for PD-L1 in non-inflamed white matter, with an induction associated with inflammation. PD-1 on CD8[+] $T_{RM}$ cells may support CNS homeostasis by preventing uncontrolled T-cell reactivity, which is a risk factor in (auto)inflammatory conditions, including multiple sclerosis[57].

CTLA-4 is expressed by both CD4[+] and CD8[+] T cells following activation and has as main function the regulation of CD28 signaling[35]. In murine brain-derived CD103[+] CD8[+] $T_{RM}$ cells, an upregulation of CTLA-4 has been reported earlier[11]. Downregulation of CD28 with high expression of CTLA-4 reflects a profound inhibition of this co-stimulatory pathway, which may be instrumental in the tight control of T cell-mediated inflammation in the brain parenchyma. Interestingly, treatment of patients with the CTLA-4 immune checkpoint inhibitor ipilimumab has been associated with the occurrence of inflammatory demyelinating white matter lesions with T-cell infiltrates[58–60]. Immunostaining for CD86 is also induced in glia in an inflammatory active MS lesion and absent in non-inflamed tissue. Likewise, mouse microglia did not express CD86 when analyzed directly ex vivo, however this was spontaneously upregulated after culture[61]. This suggests a microenvironment in situ in which expression of CTLA-4's main ligand is also tightly regulated.

Brain CD4[+] T cells also display $T_{RM}$-cell surface markers previously identified on CD4[+] $T_{RM}$ cells in other tissues[22,29]. Historically, CD8[+] T cells were regarded as the key-players in antiviral immunity, with CD4[+] T cells being mediators of the adaptive immune response against extracellular pathogens. In recent years, this dogma has been challenged by several studies: CD4[+] T cells facilitate recruitment of other lymphocytes into lymph nodes or sites of infection, provide help to CD8[+] T cells and antibody-producing B cells, and offer direct effector function through production of cytokines or lytic enzymes[62]. Although surface markers on brain CD4[+] T cells suggest a cytotoxic phenotype, cytotoxic capacity is restricted. Granzyme B and perforin are highly neurotoxic, while granzyme K by itself mediates limited cytotoxicity and rather acts as proinflammatory signal[63]. The functional role of CD4[+] $T_{RM}$ cells in the CNS requires further study.

The major novelty and strength of this study is the analysis of post-mortem brain-derived T cells with flow cytometry directly ex vivo. For primary human microglia isolated with a similar procedure, culture in vitro modified their phenotype[64]. Comparable effects are unlikely to bias our results. However, previous work in mice showed isolation of T cells from tissue for flow cytometry to affect quantity and phenotypic characteristics when compared to imaging in situ[18]. Furthermore, our study was not designed, and thereby neither powered nor matched, to explore differences in T-cell phenotypic profiles between individual brain diseases.

Together, we for the first time demonstrate human brain-derived T cells to harbor $T_{RM}$-cell features. Gained insights may help understanding how these cells mediate local protection and may be used to study how brain $T_{RM}$ cells contribute to neuroinflammatory and neurodegenerative diseases.

## Methods

**Brain donors**. Brain tissue samples were obtained from the Netherlands Brain Bank (NBB; www.brainbank.nl). The NBB obtained permission from the donors for brain autopsy and the use of tissue, blood, and clinical information for research purposes (ethical statement available at [www.brainbank.nl/media/uploads/file/Ethical-declaration.pdf])[65,66], and all procedures of the NBB have been approved by the Ethics Committee of VU University Medical Center (Amsterdam, The Netherlands). For the present study, eligibility criteria were: acquisition of both a PB sample and a subcortical normal-appearing white matter sample and time between death and end-of-autopsy < 12 h. Donor characteristics are provided in Table 1.

**Cell isolation**. Subcortical white matter samples in a range of 5–10 g/donor were stored at 4 °C in Hibernate A medium (Brain Bits LLC, Springfield, IL, USA) until workup. Before workup, a small tissue fragment was snap-frozen in liquid nitrogen and stored at −80 °C for immunohistochemistry. The remaining tissue was mechanically dissociated, followed by enzymatic dissociation, as we described previously[14]. After lysis of erythrocytes, mononuclear cells were separated from the suspension by Percoll gradient centrifugation. PBMCs from deceased brain donors, acquired by puncture of either the heart or the iliac artery for collection of 5 ml blood, and from anonymous blood donors (buffy coats; Sanquin Blood Supply Foundation, Amsterdam, The Netherlands) were isolated using standard density gradient centrifugation.

**Cell stimulation**. To assess intracellular cytokine production, cells were stimulated for 4 h with phorbol 12-mystrate 13-acetate (PMA)/ionomycin in the presence of brefeldin A, monensin, anti-CD28 (15E8, 2 µg/ml), and anti-CD29 (TS2/16, 1 µg/ml) and compared with a control sample, following earlier published procedures[67].

**Flow cytometry**. Cells were stained with antibodies for surface markers and with LIVE/DEAD™ Fixable Red Stain or Fixable Viability Dye eFluor 780 (both Life Technologies, Bleiswijk, The Netherlands) for 30 min at 4 °C. Subsequently, cells were washed, fixed, and permeabilized, using the Foxp3/Transcription Factor Staining Buffer Set (eBioscience, Vienna, Austria) or BD Cytofix/Cytoperm™kit (BD Biosciences, Breda, The Netherlands), followed by intracellular staining with antibodies. Washed cells were analyzed at a LSRFortessa™ cell analyzer (BD Biosciences, San Jose, CA, USA). FlowJo software (Tree Star, Ashland, OR, USA) was used for subsequent data analysis. Hierarchical stochastic neighbor embedding (HSNE) analysis was performed with Cytosplore[+HSNE] software[20]. Specifications of the used antibodies are provided in Supplementary Table 1.

**Immunohistochemistry**. Sections from formalin-fixed, paraffin-embedded (FFPE) and cryopreserved white matter tissue were used for immunohistochemistry. FFPE sections were deparaffinized, and antigen retrieval was performed in a microwave at 800 W for 10 min in citrate buffer (10 mM citric acid, pH 6). Cryosections were fixated in 4% paraformaldehyde buffer for 10 min. Endogenous peroxidase activity was blocked with 3% $H_2O_2$, 0.2% Triton-X in Tris-buffered saline. The sections were incubated overnight at 4 °C with primary antibodies directed against CD3 (ab669, 1:50; Abcam, Cambridge, UK), CD4 (ab133616, 1:200; Abcam), CD8 (ab4055, 1:500; Abcam), or laminin (ab80580, 1:100; Abcam). Alternatively, sections were incubated for 1 h at room temperature with primary antibodies directed against PD-L1 (clone 27A2, 1:100; Origene, Herford, Germany), CD86 (clone BU63 1:00; NSJ Bioreagents, San Diego, CA, USA), or CD103 (ab129202, 1:1,000; Abcam). For immunofluorescence, sections were incubated with biotinylated-anti-mouse (1:400), donkey-anti-rabbit Cy3 (1:400), or donkey-anti-rat-Cy5 (1:400) antibodies, sections were then incubated for 45 min with Streptavidin-Alexa 488. For the laminin and CD3 stainings, sections were subsequently incubated with rat-absorbed-biotinylated-anti-mouse antibody (Vector Laboratories). For all sections, a final incubation with Hoechst 1:1000 for 10 min was performed. Negative controls with discard of primary antibody were included. Pictures were taken using Leica TCS SP8 confocal microscope and Leica Application Suite X (Wetzlar, Germany) at 20× and 63× magnification. For immunohistochemistry, binding of biotinylated secondary antibody was visualized with avidin-biotin horseradish peroxidase complex (Vector Elite ABC kit; Vector Laboratories) and 3,3′-diaminobenzidine (EnVision; DAKO) as chromogen. Nuclei were counterstained using hematoxylin. Pictures were taken with a AxioScope microscope (Zeiss, Oberkochen, Germany) at 20× or 40× magnification using a MicroPublisher 5.0 camera (QImaging, Surrey, BC, Canada) and ImagePro Plus 6.3 software (Media-Cybernetics, Rockville, MD, USA).

For immunohistochemical quantification of CD4[+] and CD8[+] cells, 10 × 10 tiled bright-field pictures of FPPE sections were taken at 10× magnification. Images covered a median of 42 (IQR 39–48) mm[2] overlapping normal-appearing white matter. Images of complete cryostat sections stained for laminin and CD8 were made for proportional quantification of perivascular and parenchymal CD8[+] T cells. Cell counts were obtained using image J software[68,69].

**Statistics**. All results were analyzed with GraphPad Prism (GraphPad Software, La Jolla, CA, USA). Distribution of data in dot-plots is provided, as well as bars to indicate median values in the figures. Non-parametric statistical tests were employed. Unpaired samples were analyzed with the Mann–Whitney $U$ test, the paired samples of two strata were analyzed with the Wilcoxon Signed Ranks test, or the paired samples of three strata were assessed with the Friedman test, utilizing the Wilcoxon signed ranks as post hoc test. A $p$-value < 0.05 was considered significant.

## Data availability

The data that support the findings of this study, as well as specific flow cytometry panels, are available from the corresponding author upon reasonable request.

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

## Acknowledgements

We are grateful to the brain donors and their families for their commitment to the Netherlands Brain Bank donor program. We thank Dr. Corbert van Eden for experimental support, Dr. Klaas van Gisbergen, Prof. Hsi-Hsien Lin, and Prof. Hanspeter Pircher for providing antibodies, and the team of Netherlands Brain Bank (www. brainbank.nl) for their excellent service. This study was financed by MS Research grant MS13-832.

## Author contribution

J.S., K.M.H., and N.L.F. conducted experiments, J.S., K.M.H., N.L.F., E.B.M.R., P.H., I.J.M.t.B., R.A.W.v.L., I.H., and J.H. designed research, analyzed data, and interpreted results, J.S., K.M.H., and J.H. and wrote the manuscript.

## Additional information

**Competing interests:** The authors declare no competing interests.

