## [Peer Review File · Nature Communications]

Reviewers' comments:

Reviewer #1 (Tissue-resident immunity, CD4/CD8, memory)(Remarks to the Author):

The authors in this manuscript characterize CD8+T cells in human brains derived from warm autopsy specimens along with blood from the same donors. They identify that the majority of CD8+T cells in brain are CD69+, with both CD69+CD103+ and CD69+CD103- subsets. This pattern of CD69 upregulation and variable CD103 expression by memory CD8+T cells denotes tissue resident populations (TRM cells) and is confirmatory of previously reported findings by many groups in multiple human (and mouse) tissue sites. They examine expression of additional phenotypic markers and functions (cytokines, effector molecules) associated with brain TRM cells which are shared by other tissue TRM, and purport to identify two subsets of TRM cells based on CD103 expression. While analysis of TRM in human brain is novel, the study is lacking key analysis of brain T cells, and the conclusion of two TRM subsets is not supported by their data as there are few phenotypic or functional differences between CD103+ and CD103- brain TRM. The study does not provide key information as to the density of T cells in the brain using basic IHC or other visualization approaches, whether CD4 T cells are also present, and moreover, the brains are derived from individuals with neurological autoimmune diseases and other syndromes, and whether T cell or TRM content or organization differed is not addressed. There are also human subjects concerns. Specific comments are below.

1. Figure 1 shows phenotypic characterization of brain CD8 T cells based on different markers. Were other T cells present, like CD4 T cells? Did the T cell content (CD4 and CD8 lineage cells) differ in brains from different disease states? MS is associated with CD4-driven pathology. Were CD4 T cells present in MS brains—were they tissue resident? Fig. 1 needs more information or a legend for intensity and quantitation.
2. In figure 3, CD103 immunofluorescence in a brain section is shown, but there are no markers that stain for structures in the brain. A key question that arises given the presence of TRM in brain is where there are localized and what is their overall density in the tissue, but this wasn't done. Moreover, what is the distribution of CD103+ versus CD103- CD8 T cells in the brain? This results could provide insight into whether CD103 expression denotes a distinct functionality.
3. The functional analysis shows that brain TRM produce multiple pro-inflammatory cytokines following stimulation with PMA/ionomycin. Again—this result confirms what we know about TRM and memory T cells in general. The difference in cytokine levels between CD103+ and CD103- subsets is seen with only a few cytokines, and it is not clear if these minor quantitative differences define CD103 expression as denoting a distinct subset.
4. The human subjects section states that individuals were consented when alive to donate blood, brain and tissue samples-- except 6 of the patients died by euthanasia, or assisted suicide. When were they consented? Was this decision to donate tissues for research made in conjunction with their decision to end their lives? How were they euthanized?

Reviewer #2 (CNS path, DC immunity)(Remarks to the Author):

Tissue resident memory T cells in the brain might represent a critical cell population that provides the first line of defense in the CNS. In spite of their importance, there is only limited information about the phenotype and function of these cells in humans. In a previous 2013 Acta Neuropathology publication, the authors described some characteristics of differentiated CD8+ and CD4+ T cells in the human brain. In this follow-up work, the authors further describe the characterization of tissue resident memory CD8 T cells isolated from human brain samples. While the definition of two new

subpopulations of human brain TRM cells brings novel information, in the absence of correlation with different diseases states, the significance of these findings remains in question. The authors studied 17 diseased brain samples from AD, MS, PD and FTD patients and 1 sample from a patient with metastasized carcinoma without detectable brain disease. Biological variables, such as ages also varied between 47-90, and sexes were mixed. There are numerous exciting questions that could have been addressed utilizing more human brain samples from the Netherland Brain Bank. The correlation of CD8 TRM with CNS disease stages, or donor ages would lead to critical novel knowledge regarding their function, and responsiveness to different inflammatory microenvironment within the CNS. In other tissues, it had been described that TRM cells display a distinct gene expression profile that is influenced by the tissue environment, and different from naïve or TCM or TEM cells. The possibility that TRM cells are chronically activated in some CNS diseases and contribute to disease pathogenesis should also be taken into consideration. Altogether, this is an interesting, but descriptive paper that shows human brain CD8 T cell phenotypes that are similar to TRM described in different mouse disease models.

Additional comments:

1. FigS2 – Differences between CD45RO and CD28 levels in “alive” and “deceased” samples show that the expression levels of these molecules are different between these groups. Further explanation would be necessary in order to support standard comparable quality for “deceased” samples.
2. Fig 1. Sample sizes for statistics should be listed in figure legend. It is unclear how many independent analyses has been done.
3. Fig 2. Quantification for some of the small % samples is difficult to evaluate (for example GPR56). In the absence of clear statement indicating sample sizes for each staining, data presentation is unclear.
4. Fig 3I, J would need further labeling and explanation. They indicate that borders of the perivascular space were designated based on histological hallmarks, and are marked with a dotted-white line, however, “histological hallmarks” are not discussed. Additionally, exact localization of parenchymal and perivascular areas within the brain sections should be detailed. “N” and “M” are indicated in the legend but unmarked on the figure.
5. Fig 4C. What is the reason for high variability of eomes expression in the double negative population?
6. They indicate that a subset of CD8 T cells produced GM-CSF (Fig 5 M, O), however in all of these assays, GM-CSF intracellular expression was measured without quantification of cytokine production. Similarly, IFN-g or TNF-a expression was measured without measuring cytokine levels. Additionally, GM-CSF expression levels were minimal and highly variable between samples for statistical conclusions.

Reviewer #3 (Neuro-immune crosstalk, MS)(Remarks to the Author):

This manuscript reports a study of brain-isolated CD8⁺ T cells to characterise their phenotypic and functional profile. Through a detailed multiparameter flow-cytometric characterisation, the authors demonstrate that CD8⁺ T cells from human brain bear T resident memory (Trm) features and contain two subsets of CD103⁻CD69⁺ and CD103⁺CD69⁺ T cells. CD103⁺ cells display a distinctive effector molecule production profile, with production of GM-CSF and increased polyfunctionality compared to their CD103⁻ counterpart. The 103⁺ cells express high levels of PD-1 and CTLA-4, inhibitory molecules of cytotoxicity.

This paper has a number of strengths that include the inclusion of a fair number of brain donors with short (<12 hrs) postmortem duration, and thorough multiparameter flow work with good strategy for

data analysis.

To the authors' and to my knowledge, this is the first report of an in-depth characterization of CNS-derived CD8+ T cells enabling their characterization as Trm cells.

It would be important to know if the authors also analysed CD4 T cells but only chose to present data on CD8 cells. I cannot imagine that they would go through the trouble of performing the whole study without looking at CD4 cells.

Authors are encouraged to discuss the limitations and potential pitfalls of a study relying on the extraction of live T cells from postmortem tissue, with special regard to risk of selective loss of cell subpopulation and phenotypic as well as functional alterations induced by the isolation procedure to the cells that are recovered.

The results are not always presented in a way that is easy to understand. For example, the legend to Fig. 1 is insufficient to understand the figure. The colour coding should be described. Why are the scatter plot for blood and brain represented using blue and red, respectively, and the cluster plots on the right hand side use blue and yellow colours?

More attention should be given to enabling others to reproduce or extend the results from the study. To that end, the authors should publish not only the list of antibodies as they did in Table S1, but also show how they were arranged in specific staining panels.

Minor points:

There is a small number of typos to be amended. Two examples:

In ABSTRACT:

Neurotropic not neurotrophic

In Table 1: mammary (not mamma) carcinoma

Comments to reviewer #1

“The authors in this manuscript characterize CD8+T cells in human brains derived from warm autopsy specimens along with blood from the same donors. They identify that the majority of CD8+T cells in brain are CD69+, with both CD69+CD103+ and CD69+CD103- subsets. This pattern of CD69 upregulation and variable CD103 expression by memory CD8+T cells denotes tissue resident populations (TRM cells) and is confirmatory of previously reported findings by many groups in multiple human (and mouse) tissue sites. They examine expression of additional phenotypic markers and functions (cytokines, effector molecules) associated with brain TRM cells which are shared by other tissue TRM, and purport to identify two subsets of TRM cells based on CD103 expression. While analysis of TRM in human brain is novel, the study is lacking key analysis of brain T cells, and the conclusion of two TRM subsets is not supported by their data as there are few phenotypic or functional differences between CD103+ and CD103- brain TRM. The study does not provide key information as to the density of T cells in the brain using basic IHC or other visualization approaches, whether CD4 T cells are also present, and moreover, the brains are derived from individuals with neurological autoimmune diseases and other syndromes, and whether T cell or TRM content or organization differed is not addressed. There are also human subjects concerns. Specific comments are below.

Response to general remarks

We thank the reviewer for his or hers careful analysis of our work. We agree that our work is largely confirmatory of T_{RM} phenotypes earlier described in animal brains and other human tissues. We think this reproducibility stresses the validity of our findings and underlines our pioneering and novel approach of analyzing rapid post-mortem autopsy brain-derived lymphocytes with flow cytometry and even performing functional experiments. Please, find our specific responses to the reviewer's specific comments below.

Comment 1

Figure 1 shows phenotypic characterization of brain CD8 T cells based on different markers. Were other T cells present, like CD4 T cells? Did the T cell content (CD4 and CD8 lineage cells) differ in brains from different disease states? MS is associated with CD4-driven pathology. Were CD4 T cells present in MS brains—were they tissue resident? Fig. 1 needs more information or a legend for intensity and quantitation.

Response to comment 1

This comment deals with several important issues.

1. Since the majority of work on brain T_{RM} cells in mice has been conducted in $CD8^+$ T cells, and these cells also make up the majority of human brain T cells, we initially focused our manuscript on this cell fraction. Also, the flow cytometry panels of our study were primarily designed to assess $CD8^+$ T_{RM} cells. Of course, we also included a CD4 antibody in our panels. As requested by all 3 reviewers, we analyzed the expression of T_{RM} -related molecules on brain $CD4^+$ T cells.

We performed additional quantification of $CD4^+$ and $CD8^+$ cells with flow cytometry and immunohistochemistry (Figure 1), explored the $CD4^+$ T cell general phenotypic profile (Figure S5),

and analyzed expression of T_{RM} cell-associated markers on $CD4^+$ T cells (Figure 8 and S6). These new analyses are highlighted in the Abstract, Introduction (page 4), Methods (page 8), Results (page 11-12), and Discussion (page 15) sections. In our analysis, $CD4^+$ T cells also express T_{RM} markers. However, there is a notable lower expression of CD103, which is also found in $CD4^+$ T_{RM} cells in other tissues.

2. We did not focus on differences between individual diseased states. The study was designed to provide a detailed phenotypic profile of brain T cells in general. It lacks power, and no matching was performed to allow additional analyses in relation to brain pathology.

The reviewer asked specifically for donors with multiple sclerosis: if we compare sex-ratio, age, and post-mortem delay, these donors are not matched by any means with the other donors (please, see the table below with median values are shown were appropriate). This makes any differences between strata difficult to interpret, since numbers of subgroups are very small.

n/median	Non-MS	MS
Sex (M/F)	5/7	¼
Age (yrs)	77.5	50
PMD (h:m)	6:08	9:45
PH CSF	6.4	6.5

If we focus on the MS donors and compare them with the non-MS donors, no significant differences could be observed in exploratory analyses (please, see the figure below). However, due to the precautions discussed above, such statement should be interpreted with care.

MS and non-MS normal-appearing white matter $CD4^+$ and $CD8^+$ T cells express CD69 and CD103

MS and non-MS normal-appearing white matter CD69⁺ T cells express core-phenotypic T_{RM} cell markers

*In summary, in order to prevent unsubstantiated conclusions from entering scientific literature, we opted in our manuscript not to speculate on small differences between disease states, but rather focus on the general phenotypic profile of brain T cells. Although some variation exists, this profile is quite similar between all disease states. Given our earlier work, it is clear that we are particularly interested in the disease process of MS. Our current dataset can be regarded as a framework, from which we design other studies to make comparisons between MS and control donors, and to compare MS lesions with MS normal-appearing white matter. **We now discuss this obvious limitation in the Discussion section (page 15).***

*3. All reviewers missed data in the legend of Figure 1 (original version of manuscript). **The legend of Figure 2 (revised manuscript) was amended with the requested information.***

Comment 2

In Figure 3, CD103 immunofluorescence in a brain section is shown, but there are no markers that stain for structures in the brain. A key question that arises given the presence of TRM in brain is where there are localized and what is their overall density in the tissue, but this wasn't done. Moreover, what is the distribution of CD103⁺ versus CD103⁻ CD8 T cells in the brain? This results could provide insight into whether CD103 expression denotes a distinct functionality.

Response to comment 2

The reviewer poses very relevant questions, which we indeed addressed only partly in the original version of our manuscript.

*1. Regarding the overall density of CD4⁺ and CD8⁺ T cells in white matter, we performed additional immunohistochemical experiments, in which we quantified the presence of CD4⁺ and CD8⁺ lymphocytes in adjacent normal white matter sections of n=5 donors. These results are now provided in **Figure 1 (revised manuscript)**.*

2. In Figure 3 (original version manuscript), we showed examples of CD103⁺ and CD103⁻T-cell stainings in parenchyma and perivascular space, in which these compartments were identified based on the close relationship of cells with the extraluminal side of blood vessels (which can also a

appreciated in *Figure 1D and 1F (revised manuscript)*. **We specified in the legend of *Figure 4 (revised manuscript)* the hallmarks used for this identification. To further substantiate this point, we quantified CD8⁺ T-cell localization in association with laminin staining to identify cells in the perivascular space, when compared to the brain parenchyma, in cryostate section of n=4 donors. We present these results in *Figure 1 (revised manuscript)*. Only about 5% of CD8⁺ T cells is encountered in the brain parenchyma. Therefore, the substantial larger CD103⁺ fraction does not reflect cells exclusively from the brain parenchyma. This is further substantiated by the stainings in *Figure 4 (revised version manuscript)*, showing CD103⁺ cells both in perivascular space and parenchyma. To further elaborate on these staining, we performed triple-stainings of CD103, CD3, and laminin, and added a supplemental figure showing a CD103⁺ T cell within the perivascular space (figure S4).**

Comment 3

The functional analysis shows that brain T_{RM} cells produce multiple pro-inflammatory cytokines following stimulation with PMA/ionomycin. Again—this result confirms what we know about TRM and memory T cells in general. The difference in cytokine levels between CD103⁺ and CD103⁻ subsets is seen with only a few cytokines, and it is not clear if these minor quantitative differences define CD103 expression as denoting a distinct subset.

Response to comment 3

The general differences in differentiation markers, chemokine receptors/integrins, transcription factors, cytokines, and, most notably, lytic enzyme expression between CD69⁺CD103⁻ and CD69⁺CD103⁺ CD8⁺ T cells in the human brain are supportive of a different functional profile of these cells. We agree with the reviewer that these cells may be part of a continuum, and that we did not fully identify further denominators of these distinct phenotypes. We added some nuance to our claim of two distinct subsets (please, note the changed Title and Discussion section (page 13)) and acknowledge that more work needs to be done to fully understand the meaning of CD103-positivity of CD8⁺ brain T cells.

Comment 4

The human subjects section states that individuals were consented when alive to donate blood, brain and tissue samples-- except 6 of the patients died by euthanasia, or assisted suicide. When were they consented? Was this decision to donate tissues for research made in conjunction with their decision to end their lives? How were they euthanized?"

Response to comment 4

*Euthanasia or physician-assisted suicide (EAS) is legally permitted in the Netherlands under the Termination of Life on Request and Assisted Suicide (Review Procedures) Act of 2002. All cases of euthanasia included in our study were conducted within this framework. To make this clear to the readers of our manuscript, we added 'legal euthanasia' to **Table 1** and explain its meaning in the footnotes of this table. In the participants of our study, euthanasia was conducted according to*

legally obligated procedures with a sedative (pentothal/ pentobarbital/ thiopental) and muscle relaxant (pancuronium bromide/ rocuronium bromide).

*The donor program of NBB consents participants often many years prior to death and adheres to a strict ethical code of conduct, which has been published (Klioueva et al., Journal of Neural Transmission 2015; Klioueva et al., Handbook of Clinical Neurology 2018) and is available online (<https://www.brainbank.nl/media/uploads/file/Ethical-declaration.pdf>). **References to these sources are now added to the revised manuscript (page 5).***

Registration as a brain donor at the Netherlands Brainbank is never in conjunction with euthanasia, although donors occasionally register during the last few months of their lives. Comparison between donors who died from euthanasia vs. non-euthanasia does not give any evidence of a shorter time between registration and death (median 5 yrs (IQR 1–7.5, min–max 0–22) for euthanasia vs. 3 yrs (IQR 1–8, min–max 0–22) for non-euthanasia).

Comments to reviewer #2

Tissue resident memory T cells in the brain might represent a critical cell population that provides the first line of defense in the CNS. In spite of their importance, there is only limited information about the phenotype and function of these cells in humans. In a previous 2013 Acta Neuropathology publication, the authors described some characteristics of differentiated CD8⁺ and CD4⁺ T cells in the human brain. In this follow-up work, the authors further describe the characterization of tissue resident memory CD8 T cells isolated from human brain samples. While the definition of two new subpopulations of human brain TRM cells brings novel information, in the absence of correlation with different diseases states, the significance of these findings remains in question. The authors studied 17 diseased brain samples from AD, MS, PD and FTD patients and 1 sample from a patient with metastasized carcinoma without detectable brain disease. Biological variables, such as ages also varied between 47-90, and sexes were mixed. There are numerous exciting questions that could have been addressed utilizing more human brain samples from the Netherland Brain Bank. The correlation of CD8 TRM with CNS disease stages, or donor ages would lead to critical novel knowledge regarding their function, and responsiveness to different inflammatory microenvironment within the CNS. In other tissues, it had been described that TRM cells display a distinct gene expression profile that is influenced by the tissue environment, and different from naïve or TCM or TEM cells. The possibility that TRM cells are chronically activated in some CNS diseases and contribute to disease pathogenesis should also be taken into consideration. Altogether, this is an interesting, but descriptive paper that shows human brain CD8 T cell phenotypes that are similar to TRM described in different mouse disease models.

Response to general remarks

*Thank you for your critical assessment of our work. We agree that our dataset provides an initial description of T_{RM} cells isolated from human brain tissue and lacks by design matching and power to compare differences between individual disease groups. **We refer for a detailed rationale for not comparing individual disease groups to our response to comment 1 of reviewer #1. Since we agree with reviewer 2 that this is a clear limitation of our dataset, we now discuss this in the Discussion Section (page 15).***

Comment 1

1. FigS2 – Differences between CD45RO and CD28 levels in “alive” and “deceased” samples show that the expression levels of these molecules are different between these groups. Further explanation would be necessary in order to support standard comparable quality for “deceased” samples.

Response to comment 1

We realized that analysis of post-mortem peripheral blood mononuclear cells requires additional data for the readers, therefore we included supplementary Figure S1 in the original submission. It is important to note that post-mortem and living donors were not matched: the CD27/CD45RA co-expression analyses reveals that the deceased donors included much less naïve CD27⁺CD45RA⁺ cells and far more effector-memory CD27⁻CD45RA⁺ cells. Although we tried to overcome this issue by showing comparable profiles stratified for CD27/CD45RA subsets, this does not take away that T cells

of living donors were less antigen-experienced, compared to deceased donors. Furthermore, the effect of terminal disease may also contribute. We think that the similar general profile of CD45R0 and CD28 expression between CD27/CD45RA subsets supports the validity of our data. **We added these considerations to the supporting text with supplementary Figure S2.**

Comment 2

2. Fig 1. Sample sizes for statistics should be listed in figure legend. It is unclear how many independent analyses has been done.

Response to comment 2

*All reviewers missed data in the legend of Figure 1 (original version of manuscript). **The legend of Figure 2 (revised manuscript) was amended with the requested information.***

Comment 3

3. Fig 2. Quantification for some of the small % samples is difficult to evaluate (for example GPR56). In the absence of clear statement indicating sample sizes for each staining, data presentation is unclear.

Response to comment 3

*We agree that this information cannot be extracted from the figure panels. **We added the amount of donors from which data were collected to the legend of Figure 3 (revised manuscript).***

Comment 4

4. Fig 3I, J would need further labeling and explanation. They indicate that borders of the perivascular space were designated based on histological hallmarks, and are marked with a dotted-white line, however, "histological hallmarks" are not discussed. Additionally, exact localization of parenchymal and perivascular areas within the brain sections should be detailed. "N" and "M" are indicated in the legend but unmarked on the figure.

Response to comment 4

Thank you for your critical assessment of our figures, we corrected the labeling.

We performed additional stainings of CD8⁺ cells with laminin, to delineate the perivascular space, and added these to Figure 1 (revised manuscript). These were quantified to make the point that about 95% of CD8⁺ T cells resides in the perivascular space. Figure 4 (revised manuscript) makes the point that CD103⁺ cells were found both in the perivascular space as well as in the parenchyma. The neuropathological hallmarks on which the perivascular space is identified is mainly the close relationship of lymphocytes with the extraluminal side of vasculature (as can be appreciated in the figure 1D and 1F). We specified these hallmarks in the legend of Figure 4 (revised manuscript). Furthermore, we added a supplementary figure to support the presence of CD103⁺ T cells within the perivascular space (Figure S4 (revised manuscript).

Comment 5

5. Fig 4C. What is the reason for high variability of eomes expression in the double negative population?

Response to comment 5

We do not have a definitive answer to this question. We think the rather small CD69⁻CD103⁻ CD8⁺ T-cell fraction comprises circulating T cells, which differ in their differentiation status. This assumption is supported by the observation that KLRG1 expression levels also largely vary within these cells (Figure 3N). Expression of KLRG1 and Eomes indicates effector cells senescence versus memory formation, respectively.

Comment 6

They indicate that a subset of CD8 T cells produced GM-CSF (Fig 5 M, O), however in all of these assays, GM-CSF intracellular expression was measured without quantification of cytokine production. Similarly, IFN-g or TNF-a expression was measured without measuring cytokine levels. Additionally, GM-CSF expression levels were minimal and highly variable between samples for statistical conclusions.

Response to comment 6

*We agree that caution should be taken into account in the interpretation of the GM-CSF production data. Indeed, only positive fractions rather than protein levels were quantified. Variability is large and, although the median GM-CSF-positive proportion increased by 50%, intra-individual increases are limited. **Therefore, we attenuated our claims regarding this statistically significant yet modest increase in GM-CSF-positive proportions in the Results section (page 10-11).***

Comments to reviewer #3

This manuscript reports a study of brain-isolated CD8⁺ T cells to characterize their phenotypic and functional profile. Through a detailed multiparameter flow-cytometric characterisation, the authors demonstrate that CD8⁺ T cells from human brain bear T resident memory (Trm) features and contain two subsets of CD103⁻CD69⁺ and CD103⁺CD69⁺ T cells. CD103⁺ cells display a distinctive effector molecule production profile, with production of GM-CSF and increased polyfunctionality compared to their CD103⁻ counterpart. The 103⁺ cells express high levels of PD-1 and CTLA-4, inhibitory molecules of cytotoxicity.

This paper has a number of strengths that include the inclusion of a fair number of brain donors with short (<12 hrs) postmortem duration, and thorough multiparameter flow work with good strategy for data analysis.

To the authors' and to my knowledge, this is the first report of an in-depth characterization of CNS-derived CD8⁺ T cells enabling their characterization as Trm cells.

Response to general comments

Thank you!

Comment 1

It would be important to know if the authors also analysed CD4 T cells but only chose to present data on CD8 cells. I cannot imagine that they would go through the trouble of performing the whole study without looking at CD4 cells.

Response to comment 1

We designed our study and our staining panels primarily to analyze CD8⁺ T_{RM} cells, since these have been most extensively studied in animal models and other human tissues. Of course, we also stained CD4⁺ T cells and gathered these data.

We performed additional quantification of CD4⁺ and CD8⁺ cells with flow cytometry and immunohistochemistry (Figure 1), explored the CD4⁺ T cell general phenotypic profile (Figure S5), and analyzed expression of T_{RM} cell-associated markers on CD4⁺ T cells (Figure 8 and S6). These new analyses are highlighted in the Abstract, Introduction (page 4), Methods (page 8), Results (page 11-12), and Discussion (page 15) sections.

Comment 2

Authors are encouraged to discuss the limitations and potential pitfalls of a study relying on the extraction of live T cells from postmortem tissue, with special regard to risk of selective loss of cell subpopulation and phenotypic as well as functional alterations induced by the isolation procedure to the cells that are recovered.

Response to comment 2

The reviewer touches on several valid considerations when working with post-mortem human tissue, especially when isolating viable cells with rapid post-mortem autopsies.

With microglia, we know that culture induces a distinct phenotype. The strength of our study is that cells were analyzed directly after isolation without any *in vivo* culturing (of course except for the cytokine experiments). Regarding the loss of specific cell fractions, we are aware of studies in mice showing a different phenotype of T_{RM} as assessed with flow cytometry after isolation when compared to *in vivo* assessment. **We include comments on these specific limitation of our work in the Discussion section of our manuscript (page 15).**

Comment 3

The results are not always presented in a way that is easy to understand. For example Legend to Fig. 1 is insufficient to understand the figure. The colour coding should be described. Why are the scatter plot for blood and brain represented using blue and red, respectively, and the cluster plots on the right hand side use blue and yellow colours?

Response to comment 3

All reviewers missed data in the legend of Figure 1 (original version of manuscript). **The legend of Figure 2 (revised version of manuscript) was amended with the requested information.**

Comments 4

More attention should be given to enabling others to reproduce or extend the results from the study. To that end, the authors should publish not only the list of antibodies as they did in Table S1, but also show how they were arranged in specific staining panels.

Response to comment 4

*It is not very common to elaborate on the exact panels in flow cytometry papers, yet, we acknowledge that the flow-cytometric analysis of human post-mortem brain T cells is the most important novelty of our work. During the course of our study, there were also amendments to our staining protocols, e.g the exclusion of antibodies, which did not give a reproducible signal and were therefore excluded from the final dataset (for instance, anti-CXCR5). Therefore, an exhaustive list with all panels used would not add much useful information. **We add a statement that data including staining panels used are available upon request (page 7).***

Comments 5

Minor points:

There is a small number of typos to be amended. Two examples:

In ABSTRACT:

Neurotropic not neurotrophic

In Table 1: mammary (not mamma) carcinoma

Response to comments 5

Thank you for pointing out these typo's, we corrected them and screened the text for further typo's.

REVIEWERS' COMMENTS:

Reviewer #1 (Remarks to the Author):

The authors have addressed the main concerns, by providing additional data, analyses, and clarifications. The information provided will be of significant interest to the field.

Reviewer #2 (Remarks to the Author):

The authors responded as much as they can to previous comments. They included new data for brain tissue resident CD4 cells and emphasized the limitations of their data. This manuscript provides the first in-depth characterization of tissue resident T cells of the human brain.

Reviewer #3 (Remarks to the Author):

The authors have addressed to the best of their ability and feasibility the reviewers' comments. They have candidly and prudently admitted limitations of the work that attest to their scientific rigour. The changes made have improved the accuracy, completeness and ease of reading of the manuscript.

REVIEWERS' COMMENTS:

Reviewer #1 (Remarks to the Author):

The authors have addressed the main concerns, by providing additional data, analyses, and clarifications. The information provided will be of significant interest to the field.

R: thank you!

Reviewer #2 (Remarks to the Author):

The authors responded as much as they can to previous comments. They included new data for brain tissue resident CD4 cells and emphasized the limitations of their data. This manuscript provides the first in-depth characterization of tissue resident T cells of the human brain.

R: thank you!

Reviewer #3 (Remarks to the Author):

The authors have addressed to the best of their ability and feasibility the reviewers' comments. They have candidly and prudently admitted limitations of the work that attest to their scientific rigour. The changes made have improved the accuracy, completeness and ease of reading of the manuscript.

R: thank you!